# Soil and Foliar Zinc Biofortification of Triticale (x *Triticosecale*) under Mediterranean Conditions: Effects on Forage Yield and Quality

**DOI:** 10.3390/plants13141917

**Published:** 2024-07-11

**Authors:** Carlos García-Latorre, María Dolores Reynolds-Marzal, Saúl De la Peña-Lastra, Nuno Pinheiro, María José Poblaciones

**Affiliations:** 1Department of Agronomy and Forest Environment Engineering, University of Extremadura, Avda. Adolfo Suárez s/n, 06007 Badajoz, Spain; cgarcialn@unex.es (C.G.-L.); lolreymar@gmail.com (M.D.R.-M.); saul.delapena@gmail.com (S.D.l.P.-L.); 2National Institute for Agricultural and Veterinary Research (INIAV), Estrada de Gil Vaz, 7350-228 Elvas, Portugal; npinheiro39@gmail.com

**Keywords:** agronomic biofortification, zinc sulphate, rainfed conditions, forage production, nutritive parameters

## Abstract

Zinc (Zn) deficiency represents a significant global concern, affecting both plant and human health, particularly in regions with Zn-depleted soils. Agronomic biofortification strategies, such as the application of Zn fertilizers, offer a cost-effective approach to increase Zn levels in crops. This study aimed to assess the efficacy of soil and foliar Zn biofortification, applied as an aqueous solution of 0.5% zinc sulphate (ZnSO_4_·7H_2_O), on triticale (x *Triticosecale*) grown under Mediterranean conditions. The study was conducted over two growing seasons (2017/18 and 2018/19) in southern Spain, evaluating the effects on biomass yield; forage quality, including crude protein, Van Soest detergent fiber, organic matter digestibility, and relative forage value; and nutrient accumulation. Soil treatment consisted in the application of 50 kg of ZnSO_4_·7H_2_O ha^−1^ solely at the beginning of the first campaign to assess the residual effect on the second year. In contrast, the foliar treatment consisted of two applications of 4 kg of ZnSO_4_·7H_2_O ha^−1^ per campaign, one at the beginning of tillering and the other at the appearance of the first node. The foliar application increased the Zn content of the forage to adequate levels, while the soil application resulted in a 33% increase in biomass production, which is particularly beneficial for farmers. Overall quality was favored by the combined soil + foliar application, and no adverse antagonistic effects on other nutrients were detected. Instead, a synergistic interaction between Se and Zn was observed, which improved the efficacy of this important micronutrient for livestock and human wellbeing.

## 1. Introduction

In recent decades, agriculture has primarily focused on increasing food production to meet the demands of a growing population. This focus has led to the intensified use of high-yielding crop varieties, intensive cultivation methods, and the application of micronutrient-free fertilizers, which have contributed to the depletion of essential micronutrients, such as iron (Fe), copper (Cu), and zinc (Zn), in agricultural soils [1,2]. Moreover, the depletion of micronutrients in soil can lead to nutritional imbalances in crops, reducing their quality and potentially affecting human health [3]. Studies have shown a geographical overlap between regions characterized by Zn-deficient soils and populations experiencing Zn deficiency [4], indicating a concerning correlation between soil health and human well-being [5]. 

Zn plays a crucial role as a micronutrient in plant growth and development due to its involvement in various metabolic processes, including carbohydrate and protein metabolism [6]. Zn deficiency in plants can lead to stunted growth, chlorosis, spikelet sterility, and impaired water uptake and transport [7,8]. Approximately 3000 proteins, including those in animals and humans, contain Zn [9,10], and Zn is the only metal involved in six classes of enzyme [10,11]. Thus, deficient levels can lead to a range of physiological problems, including growth retardation, susceptibility to infectious diseases, impaired immune system function, and learning disabilities [12].

Zn deficiency affects approximately two billion people [8], making Zn one of the elements associated with what is currently referred to as “hidden hunger” [13]. This concept refers to the lack of essential micronutrients even when energy needs are met through consumption [14]. Cereals are among the most important crops for both livestock and humans, providing a significant portion of their daily energy requirements [15]. However, the major cause of Zn deficiency in humans is the high consumption of cereal-derived foods with low levels and low bioavailability of Zn [16]. 

This deficiency is directly related to the availability of Zn in the soil [17]. Nearly 50% of cereals soils worldwide are deficient in this element [18]. This is particularly evident in the semi-arid Mediterranean area in the Iberian Peninsula, where this deficiency is particularly pronounced in the southwest, where soils are often classified as deficient in Zn. Crops grown under such conditions often exhibit values below 0.5 mg Zn-DTPA kg^−1^ [19], resulting in low yields and low Zn concentrations throughout all parts of the plant [17], which also affect the expression of genes needed for stress protection [20].

Efforts to address the depletion of micronutrients in soils include agronomic biofortification strategies, where crops are enriched with essential nutrients through sustainable fertilization practices [21]. The application of Zn fertilizers can help to increase soil bioavailable Zn levels, thereby improving crop growth and yield [22]. Consequently, this technique offers an efficient and cost-effective method for the production of Zn-enriched food through the use of fertilizers such as Zn sulphate. When considering the application of Zn in agriculture, both soil and foliar applications play crucial roles in enhancing plant growth and nutrient uptake. Studies have shown that soil application can lead to enhanced Zn levels in the soil, impacting plant growth positively [23]. On the other hand, foliar application has been found to increase the Zn concentration in grains, thereby improving the overall nutrient quality of the crop [24]. Furthermore, their combination has been found to be particularly effective for both purposes, increasing growth and Zn enrichment [17,25,26]. And, to maximize its efficacy, this technique may be applied to staple foods such as cereals, which are consumed daily and globally. For animal feed, these would include fodder crops or grain crops when incorporated into feed formulations. Triticale (x *Triticosecale* Wittmack) could be of particular interest due to its good performance in stressful environments and its diversified applications [27]. Thus, this study aimed to assess the efficacy of soil and foliar Zn biofortification, as well as the interaction of both treatments, in triticale across two growing seasons (2017/18 and 2018/19), evaluating their effects on yield, quality, and nutrient accumulation.

## 2. Results

### 2.1. Zn Concentrations in Soil

The soil Zn treatment that was only applied at the beginning of the first campaign (2017/18) significantly increased the DTPA-extractable Zn concentration in the topsoil from an initial concentration of 0.30 ± 0.03 mg kg^−1^ to an average of 1.50 ± 0.25 mg kg^−1^ in soil Zn plots and up to 1.63 ± 0.28 mg kg^−1^ in the combination soil + foliar, according to the measurement obtained at the end of the second cropping year in May 2019.

### 2.2. Effect on Forage Yield and Quality

The ANOVA analysis showed that the main effect, study year, significantly influenced all the studied parameters except crude protein. However, the effect of Zn application influenced all parameters except for neutral detergent fiber (NDF), acid detergent lignin (ADL), and ash content. Finally, the interaction “study year x Zn application” only had a significant effect on ADL, ash content, and the relative forage value (RFV) (Table 1).

Regarding production, the forage yield harvested at the beginning of milky grain production in the first year (2017/18) averaged 1380 kg ha^−1^ higher than that harvested in the second year (2018/19). In terms of quality, the crude protein content (CP) of the triticale plants was, on average, 7.7%, without a significant difference between years. In addition, both the neutral detergent fiber (NDF) and acid detergent fiber (ADF) content were significantly higher in the second year, with values of 63.0% and 33.7%, respectively, compared to the first year, in which they were 52.5% and 29.2%, respectively. In contrast, both the acid detergent lignin (ADL) and ash content were significantly higher in 2017/18 (3.94% and 0.54%, respectively) than in 2018/19 (2.96 and 0.25%, respectively). Similarly, both the organic matter digestibility (ODM) and relative forage value (RFV) were significantly higher in the first year, with values 3.6 and 1.74 points higher, respectively (Table 2).

Regarding the effect of Zn application, the treatments that significantly increased the yield of triticale plants compared to the control were soil and soil + foliar, with 16,407 kg ha^−1^ and 18,515 kg ha^−1^, respectively (vs. 13,855 kg ha^−1^), representing a significant increase of 18.4% and 33.6%, respectively, and with significant differences between them (Table 3). In terms of quality, the foliar application showed the significantly highest protein content (CP), with an increase of 19.5% compared to no-Zn, followed by soil + foliar, although without significant difference regarding it. The only fiber affected by Zn application was the acid detergent fiber (ADF). In this sense, both Zn treatments involving soil application, either alone or in combination with foliar application, significantly reduced this parameter compared to no-Zn, by 5.0% and 3.1%, respectively. Finally, RFV and ODM showed opposite variations. Both applications that included soil application (soil and soil + foliar) significantly increased the percentage of ODM but significantly decreased RFV compared to no-Zn (Table 3). Regarding the significant interactions, while for ADL, the foliar and soil + foliar applications in the second year obtained significantly lower values, for ash content, the foliar application produced the highest value in 2017/18 but the lowest in 2018/19 (without significant difference with no-Zn). For RFV, both soil Zn applications significantly reduced its value by an average of 8.2% and 7.5%, respectively, compared to the control plots (Figure 1).

### 2.3. Effect on Forage Nutritional Quality

The main effect, study year, significantly affected the content of Ca, Mg, and Zn, but not the content of Fe and Se. In addition, it also had a significant effect on the removal of Fe, Mg, and Zn, but not on the removal of Ca nor Se. Moreover, the other main effect, Zn application, significantly affected all the parameters studied except total Fe. The interaction “study year x Zn application” only significantly affected the Ca and Mg content and Ca removal (Table 1).

Regarding the year of study, in 2017/18, the concentration of Mg and Zn, as well as the removal of Fe, Mg, and Zn, showed significantly higher values (31.0%, 68.5%, 27.2%, 42.4%, and 81.4% higher, respectively) compared to the second year (2018/19). In contrast, Ca concentration presented significantly higher values for the second study year, with an increase of 18.6%. Neither the Fe nor Se content was significantly affected by the study year, showing average concentrations of 83.4 mg kg^−1^ and 94.5 µg kg^−1^, respectively (Table 2).

With regard to the nutrient content of the soil, Zn application presented a general positive effect. In this sense, the soil application of Zn significantly increased the levels of Ca, Mg, and Se when compared to no-Zn (by 16.1%, 24.0%, and 195.7%, respectively) (Table 3). In the case of total Zn, the trend was foliar > soil + foliar > soil = no-Zn, with increases of the first two of 2.75-fold and 2.16-fold, respectively, in comparison to no-Zn. All the Zn applications significantly increased the total content of Se, with an average increase of 2.84-fold and no significant differences between treatments. The same trend can be observed for nutrient removal. The soil application of Zn significantly increased all these parameters in comparison with no-Zn, with the greatest increases observed for Ca (28.7%), Fe (49.8%), Mg (47.1%), and Zn (57.0%). Both foliar and soil + foliar significantly increased Zn removal (both treatments showed significantly (88.5%) higher values compared to soil application). Regarding Se removal, all Zn treatments exhibited a significant increase in value in comparison to no-Zn, with soil and soil + foliar demonstrating the greatest increases, exceeding a 3.5-fold increase, when compared to the no-Zn control.

The interaction “study year x Zn application” had a significant effect on Ca and Mg concentrations and Ca removal in plants (Figure 1). In 2017/18, the soil Zn application significantly increased the Ca content, while the soil + foliar application resulted in a significant reduction. In 2018/19, only the foliar treatment caused a significant decrease in the accumulation of Ca (compared to no-Zn). The soil application of Zn during the first year of the study resulted in the significantly highest value for Ca removal, improving the results of the respective no-Zn control, while the soil + foliar application during the first year also resulted in a significant reduction. Regarding total Mg, the trend was similar; while in the first year, the soil application of Zn significantly increased Mg accumulation, the soil + foliar application of Zn decreased it compared to no-Zn. During the second year, no significant difference was found between the treatments in regards to total Mg (Figure 1).

## 3. Discussion

The aim of this study was to assess the potential effect of Zn biofortification on triticale (x *Triticosecale* Wittmack) forage in areas where crop soils are typically deficient, such as the southwestern region of Spain. In this context, the initial soil content of only 0.30 mg Zn-DTPA kg^−1^ could be classified as Zn deficient, since soils with less than 0.5 mg Zn kg^−1^ are generally classified as such [28,29]. The fivefold increase in DTPA-extractable Zn concentration in the topsoil after soil Zn application is supported by the positive correlation between Zn application and its bioavailability in the soil [30,31]. Furthermore, previous studies have already reported that the combined application of soil and foliar Zn is an effective method to improve not only soil Zn content but also plant Zn concentrations in crops as diverse as wheat [32] and broccoli [33]. This outcome confirmed that the unique soil application of Zn at the beginning of the first campaign (either alone, or in combination with foliar application) could have had a residual effect during (at least) the following year. However, considering the complex relationship between soil properties, Zn application, and plant uptake, further research is needed to assess the duration of this residual effect over a longer period.

The differences in the production and quality parameters of the triticale forage harvested at the beginning of milky grain production between the two study years could be mainly attributed to the significant variation in rainfall patterns. The first year (2017/18) experienced 60% more precipitation than the second year (2018/19). Furthermore, the total spring precipitation in the first year was almost equal to all the precipitation in the second year, which had significant dry periods between February and March. Nutrient accumulation and performance in triticale have been linked to precipitation levels, emphasizing the importance of water availability for optimal yield and quality [34]. Therefore, these contrasting weather conditions likely influenced not only biomass production, with almost 1400 kg ha^−1^ more, but also the nutrient content, digestibility, and fiber quality of triticale forage, resulting in better results in these aspects during the first year. However, the higher amount of forage yield produced in the first year did not produce a lower protein content due the dilution of photosynthates because of the higher rainfall [35]. Furthermore, the better performance during the first year of study, especially the higher Zn uptake, could also be partially attributed to the Zn applied to the soil at the beginning of the first year, since this treatment was not repeated during the second year. This situation would have led to an increase in the bioavailability of Zn, from which the first triticale harvest would have benefited and whose effect could have been diluted in the second year of study. This could help explain why the protein content was not reduced during the first year due to increased production and excess rainfall, since Zn is crucial for the production of a wide range of proteins [9,10].

Regarding the nutritive quality of the triticale forage in the non-fertilized plots, the total contents found (2658 mg kg^−1^ of Ca, 77.8 mg kg^−1^ of Fe, 849 mg kg^−1^ of Mg, 8.0 mg kg^−1^ of Zn, and 39.7 µg kg^−1^ of Se) were, in general, low or very low in comparison with the generally recommended levels (2000–8000 mg kg^−1^ of Ca, 1200–1800 mg kg^−1^ of Mg, 20–30 mg kg^−1^ of Zn, and 100–200 µg kg^−1^ of Se). Only in terms of Fe can the triticale be considered rich, because the recommendations are 30–50 mg kg^−1^ [36,37,38]. These low values may be attributed to the low bioavailability of these nutrients in the soils, being the major factor affecting their accumulation in the edible parts of plants. Therefore, these soil conditions, which are quite frequent in Spain [19,39] and other parts of the world [40,41], are very appropriate for the evaluation of the suitability of forage triticale to be included in biofortification programs.

Regarding nutrient mobilization as a function of the study year, the first year led to significantly higher Zn and Mg contents and a lower Ca concentration. The higher Zn content could be attributed to the combined effect of the initial soil application and the increased availability of water, as this nutrient enters plants primarily via root absorption of Zn^2+^ from the soil solution [42], and, additionally, the higher rainfall may have facilitated its uptake and translocation within the plant via foliar application [43]. The higher rainfall could help explain the results for the higher Mg content, as the increased water-soluble Mg in the soil resulting from higher rainfall could have improved its mobility [43], leading to a greater uptake by the plants. Consequently, this could explain the significantly higher removal of Mg from the soil in the first year. In addition, the lower Ca content could be related to the antagonistic relationship that both nutrients seem to have within the plant [44]. Furthermore, the lack of a significant effect of the study year or Zn application on Fe and Se content and removal indicates that these elements may be less affected by external factors in this experimental design [16]. Thus, irregularity in precipitation, characteristic of Mediterranean conditions, can lead to inconsistencies in nutrient uptake and subsequent grain nutrient accumulation. Consequently, under such conditions, targeted fertilization, including essential minerals, could play a crucial role in compensating for the lower nutrient accumulation observed during less favorable climatic years. 

Considering the effect of the Zn biofortification treatments on forage yield, while soil application produced an increase of 18.4%, the combined soil + foliar application resulted in a significant yield increase of 33.6% (both compared to the control plots). This outcome is likely due to the synergistic effect of combining the two application methods. The soil application of Zn may have provided a baseline nutrient level for plant growth, while the foliar application provided a rapid uptake and utilization of Zn through the leaves. This combined approach resulted in a more balanced and readily available supply of Zn, ultimately improving growth and yield [45]. In addition, foliar treatment alone also showed a positive tendency, although not significant, to increase forage yield compared to the control.

In addition to its effect on improving triticale biomass, the study emphasizes the importance of the foliar application of Zn [46], which significantly increased protein content (41.4%), which has been associated with the involvement of Zn in protein synthesis [47], and, along with the improvement of forage yield, could be of vital importance in motivating farmers to engage in the implementation of these biofortification programs. Triticale is a nutritious grain with high protein, amino acid, and fiber contents, beneficial for both human and animal well-being, and it may also have preventive effects against oxidative damage [48,49]. 

Foliar application was the only treatment that achieved a Zn content higher than 20 mg kg^−1^, considered as adequate for animal requirements [38]. However, the soil + foliar combination reached 17.3 mg kg^−1^. This may be due to the weather, as the particularly dry spring of the second year prevented not only better development and production of the crop but also the correct assimilation of Zn. To correct the possible errors due to the difference in yields, the Zn removal, in which the Zn content is multiplied by the yield, shows how both foliar and soil + foliar treatments were significantly better at using Zn than the control, multiplying its value by an average of 2.96-folds. This should be considered when choosing which treatment is more interesting, as the farmer will prefer a treatment that also improves yield.

A very positive aspect is the synergic effect that all of the applications studied had on the accumulation of Se, almost tripling its value in all cases and reaching the target level of 100 µg kg^−1^ [36,38], although the soil + foliar application stands out in terms of Se removal. This synergistic effect has already been reported by numerous authors in different crops, including bread wheat [50] and forage peas [51]. The study by Reynolds-Marzal et al. [51] is of particular importance, as it conducted a biofortification trial assessing the separate and combined applications of Se and Zn. It was observed that the Zn application, whether applied alone or in combination with Se, resulted in significant increases in both the Zn and Se content of the plants. In contrast, the application of Se significantly increased the Se content but did not correspondingly increase the Zn concentration. These results highlight that Zn application may influence the uptake or assimilation pathways of both Zn and Se in plants, possibly through interactions with shared metabolic processes or transport mechanisms. Conversely, Se application primarily enhances Se uptake or assimilation pathways without similar effects on Zn uptake pathways. In this context, Se is typically assimilated in plants through a sulfur pathway [52], and the application of Zn has been observed to increase the expression of sulfate transporters [53], which could explain the effect of Zn biofortification on Se uptake.

Soil application increased the plant uptake of Ca and Mg, but combined soil + foliar treatment decreased the Ca content compared to the control. However, even in the worst case, the content was enough to accomplish the livestock requirement of 2000 mg kg^−1^ [37]. These findings are consistent with other studies [54,55,56] and support the fact that Zn biofortification strategies, whether through foliar, soil, or combined applications, play a crucial role in improving the quality and micronutrient uptake of forage crops. Iron was the only one of the nutrients studied that was not affected by the application of Zn, its content in all cases being higher than the limits established as sufficient for the correct intake of Fe in cattle (30 to 50 mg kg^−1^ [38]).

## 4. Materials and Methods

### 4.1. Study Site

The study was conducted in the growing seasons 2017/18 and 2018/19 in Badajoz, southern Spain (38°54′ N, 6°44′ W, 186 m above sea level), in a xerofluvent soil under rainfed Mediterranean conditions. The climate data collection and soil sampling of the study area are described in detail in [15]. It is important to note that the two growing seasons were substantially different. Although the rainfall of 2017/18 could be considered as normal (477 mm), there were two months, March and April, with a very high amount of rain (252 mm), which favored good crop development. The second year was especially dry, with a total rainfall of 295 mm, 35% lower than in a normal year, with a significant water shortage in February and March.

The soil, classified as xerofluvent type [57], had a clay-loam texture, slightly acidic pH (6.4 ± 0.02, mean ± standard error), low organic matter content (1.31 ± 0.09%) and extractable Zn or DTPA (0.30 ± 0.03 mg kg^−1^), and very slightly saline electrical conductivity (1.32 ± 0.02 mS cm^−1^). The soil presented with normal contents of total N (0.12 ± 0.01%), determined by the Kjeldahl method (Kjeltec™ 8200 Auto Distillation Unit. FOSS Analytical. Hilleroed, Denmark) [58], Olsen P (4.9 g kg^−1^), and extractable K (0.82 meq 100 g^−1^), which was extracted with ammonium acetate (1 N) and quantified by atomic absorption spectrophotometry (Helyos alpha, 9423-UVA, Unicam, Cambridge, UK), and a very high content of extractable Mg (3.72 meq 100 g^−1^). Ca, Mg, Fe, and Zn were extracted with DTPA (diethylenetriaminepentaacetic acid), and all of them were determined by an inductively coupled plasma mass spectrometer (ICP-MS; Agilent 7500ce, Agilent Technologies, Palo Alto, CA, USA). All the results were reported on a dry weight basis.

### 4.2. Experimental Design

The experiment design was a split-split plot with four repetitions conducted during the study years 2017/18 and 2018/19, similar to that described in [15,51] but using triticale as target crop. Plot size for each treatment was 15 m^2^ (3 m × 5 m). Soil Zn application (subplot factor), with and without 50 kg ZnSO_4_·7H_2_O ha^−1^, which was sprayed to the soil surface and then incorporated into the soil before sowing. The sub-subplot studied the foliar application (with and without application) of two foliar Zn applications at a dose of 4 kg ZnSO_4_·7H_2_O ha^−1^, one at the beginning of tillering and the other at the first node appearance. At each foliar application, an aqueous solution 0.5% (*w*/*v*) of ZnSO_4_·7H_2_O ha^−1^ with 800 L ha^−1^ was sprayed in the very late afternoon until most of the leaves were covered. Soil treatment was only made at the beginning of the first year of the experiment to evaluate its residual effect along the essay’s duration. A diagram of the experimental design is shown in Table 4.

### 4.3. Crop Management

Triticale cultivar used in the experiments was ‘Bondadoso’, with a short-medium cycle. Conventional tillage was carried out to prepare a proper seedbed before sowing. The sowing dose was 350 seeds m^−2^, and the dates were early November 2017 and late December 2018 (the intense rainfall in November 2018 did not allow earlier sowing). An N-P-K fertilizer (15-15-15) was applied before sowing at a 200 kg ha^−1^ dose in all the plots, and 100 kg N ha^−1^ was applied as urea (46%) at tillering. The dose and form of the NPK fertilizer application were based on the local common crop management. A pre-emergence herbicide (clortoluron 40% p/v + diflufenican (DFF) 2.5% p/v) was applied at a rate of 2.5 L product ha^–1^ (Harpo-Z, Bayer Crop Science S.L., Barcelona, Spain).

### 4.4. Soil and Forage Analysis

To evaluate the residual effect of the Zn application in the soil, an additional sampling (with their corresponding extractable Zn determination) was performed at harvest time in the second growing season.

Forage was harvested at the beginning of milky grain production. The harvested forage was oven-dried at 65 °C and weighed to obtain the dry forage biomass. Protein content (%) was measured by the Kjeldhal method (Kjeltec™ 8200 Auto Distillation Unit. FOSS Analytical. Hilleroed, Denmark) and used to estimate crude protein (CP) by multiplying N × 6.25 [59]. Official procedures [60] were followed to determine neutral detergent fiber (NDF), acid detergent fiber (ADF), and acid detergent lignin (ADL) by means of a fiber analyzer (ANKOM8–98, ANKOM Technology, Macedon, NY, USA). Total ash content was determined by ignition of the sample in a muffle furnace at 600 °C, as indicated by the official procedure [60]. The relative forage value (RFV) of the dry matter and the organic matter digestibility (OMD) of the forage were calculated by following the procedures proposed by [61] and [62], respectively. Total forage Ca, Fe, Mg, Se, and Zn concentrations were determined as described in [51]. Briefly, samples of each treatment were finely ground (<0.45 mm); 0.5 g was digested with ultra-pure concentrated nitric acid (2 mL) and 30% *w*/*v* hydrogen peroxide (2 mL) using a closed-vessel microwave and diluted to 25 mL with ultra-purified water [63]. For quality assurance, a blank and a standard reference material (tomato leaf, NIST 1573a) were included in each batch of samples. The nutrient-specific recovery was 94% compared with certified reference material values. Concentrations of Ca, Fe, Mg, Se, and Zn were determined by ICP-MS as described above for soil samples. To consider the dilution effect, total content of each mineral per ha was also determined by multiplying its concentration by forage yield. All results were referenced to dry matter content.

### 4.5. Statistical Analysis

All data were evaluated first for normality and homogeneity of variances using the Kolmogorov–Smirnov test and Levene’s test, respectively. The influence of the study year (2017/18 and 2018/19), soil Zn application (0SZn and 50SZn), foliar application (0F and 8FZn), and their interactions on all data (forage yield, CP, forage fiber (ADF, NDF, and ADL), ash, OMD and RFV, and total Ca, Fe, Mg, Se, and Zn) was analysed by following mixed-design models through split-split plot ANOVA). When significant differences were found via the ANOVA, means were compared using the Fisher’s protected least significant difference (LSD) test at *p* ≤ 0.05. All the analyses were performed with the Statistix v. 8.10 (2) package.

## 5. Conclusions

The current study demonstrated that triticale is well suited for an agronomic biofortification program with Zn under Mediterranean conditions, effectively increasing its concentration in forage and mitigating Zn deficiency in livestock and humans without compromising forage quality or yield. The foliar application of Zn at a rate of 8 kg zinc sulfate ha^−1^ was adequate to elevate Zn concentrations in the forage above recommended levels. Additionally, also applying 50 kg zinc sulfate ha^−1^ to the soil prior to sowing not only boosted mineral accumulation but also increased forage yield by 33%, which is particularly beneficial for farmers. No adverse antagonistic effects on other nutrients were detected; instead, a synergistic interaction between Zn and Se was observed, improving this important micronutrient for livestock and humans.

## Figures and Tables

**Figure 1 plants-13-01917-f001:**
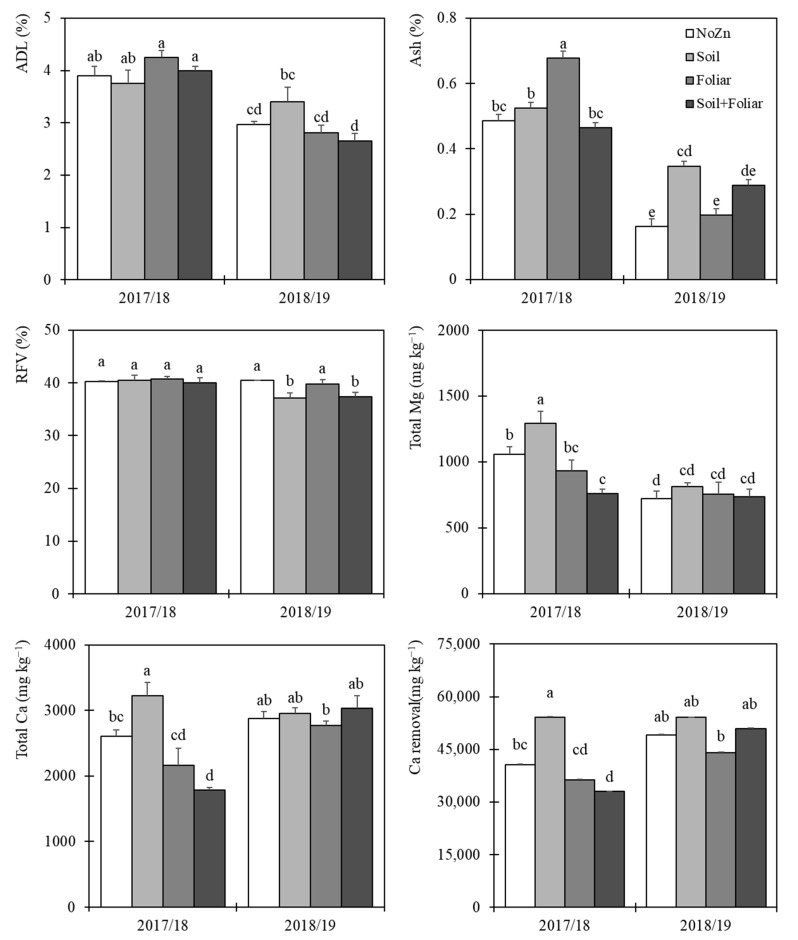
Effect of study year and Zn application interaction on acid detergent lignin (ADL), ash content, forage relative value (RV), total Mg, and Ca concentrations and Ca removal (represented by means and standard error). Different letters indicate significant differences (*p* ≤ 0.05) according to LSD test.

**Table 1 plants-13-01917-t001:** ANOVA table showing the influence of study year, Zn application, and their interactions on each parameter evaluated (n = 4).

Parameter	Year	Zinc	Year*Zinc
Yield	3.75 *	11.44 ***	0.04
CP	3.01	25.79 ***	0.63
NDF	156.81 ***	1.09	2.29
ADF	64.33 ***	3.59 *	0.57
ADL	46.66 ***	0.71	2.12 *
Ash	69.19 ***	2.15	4.36 *
OMD	67.46 ***	4.04 *	1.14
RFV	28.96 ***	7.79 ***	7.00 **
Total Ca	16.06 **	7.39 **	8.04 **
Total Fe	3.68	2.58	0.77
Total Mg	25.10 ***	7.67 **	4.23 *
Total Zn	62.81 ***	46.04 ***	2.42
Total Se	0.30	12.89 ***	1.22
Ca removal	2.94	7.38 **	4.98 *
Fe removal	6.26 *	3.32 *	0.30
Mg removal	29.14 ***	6.88 **	2.27
Zn removal	73.75 ***	48.70 ***	3.09
Se removal	0.01	15.48 ***	1.43

F-values and level of significance (* *p* ≤ 0.05, ** *p* ≤ 0.01, *** *p* ≤ 0.001) are shown for each parameter, considering both factors and their interaction. CP: crude protein; NDF: neutral detergent fiber; ADF: acid detergent fiber; ADL: acid detergent lignin; Ash: ash content; OMD: organic matter digestibility; RFV: relative forage value.

**Table 2 plants-13-01917-t002:** Effect of study year (Y) on each parameter evaluated (represented by means ± standard error).

Parameter	2017/18	2018/19
Yield (kg ha^−1^)	16,743 ± 610 A	15,360 ± 564 B
CP (%)	7.43 ± 0.4	7.94 ± 0.4
NDF (%)	52.5 ± 0.7 B	63.0 ± 0.7 A
ADF (%)	29.2 ± 0.4 B	33.7 ± 0.5 A
ADL (%)	3.94 ± 0.10 A	2.96 ± 0.13 B
Ash (%)	0.54 ± 0.03 A	0.25 ± 0.03 B
OMD (%)	66.1 ± 0.3 A	62.5 ± 0.4 B
RFV	40.4 ± 0.2 A	38.7 ± 0.5 B
Ca (mg kg^−1^)	2429 ± 189 B	2881 ± 64 A
Fe (mg kg^−1^)	89.1 ± 6.0	77.0 ± 3.8
Mg (mg kg^−1^)	990 ± 69 A	756 ± 30 B
Zn (mg kg^−1^)	18.2 ± 2.0 A	10.8 ± 1.8 B
Se (µg kg^−1^)	91.5 ± 12.0	97.4 ± 13.5
Ca removal (g ha^−1^)	40,947 ± 3160	45,130 ± 2411
Fe removal (g ha^−1^)	1521 ± 135 A	1195 ± 67 B
Mg removal (g ha^−1^)	16,716 ± 1236 A	11,766 ± 599 B
Zn removal (g ha^−1^)	312 ± 37 A	172 ± 30 B
Se removal (mg ha^−1^)	1597 ± 233	1592 ± 264

Different letters (if any) indicate significant differences (*p* ≤ 0.05) according to LSD test. CP: crude protein; NDF: neutral detergent fiber; ADF: acid detergent fiber; ADL: acid detergent lignin; Ash: ash content; OMD: organic matter digestibility; RFV: relative forage value.

**Table 3 plants-13-01917-t003:** Effect of Zn application (Zn) on each parameter evaluated (represented by means ± standard error).

Parameter	No-Zn	Zn Soil	Zn Foliar	Zn Soil + Foliar
Yield (kg ha^−1^)	13,855 ± 446 C	16,407 ± 552 B	15,430 ± 843 BC	18,515 ± 632 A
CP (%)	7.0 ± 0.2 BC	6.5 ± 0.3 C	9.9 ± 0.4 A	7.4 ± 0.3 B
NDF (%)	57.2 ± 1.9	56.9 ± 2.6	58.9 ± 1.8	57.9 ± 2.8
ADF (%)	31.9 ± 1.2 A	30.3 ± 1.0 B	32.7 ± 0.9 A	30.9 ±1.2 B
ADL (%)	3.37 ± 0.16	3.58 ± 0.15	3.53 ± 0.32	3.33 ± 0.34
Ash (%)	0.33 ± 0.06 B	0.44 ± 0.06 A	0.44 ± 0.10 A	0.38 ± 0.04 AB
OMD (%)	63.7 ± 1.0 BC	65.3 ± 0.8 A	63.5 ± 0.7 C	64.9 ± 0.9 AB
RFV	40.3 ± 0.1 A	38.8 ± 0.7 B	40.3 ± 0.5 A	38.8 ± 0.6 B
Ca (mg kg^−1^)	2658 ± 115 B	3087 ± 129 A	2468 ± 198 BC	2408 ± 321 C
Fe (mg kg^−1^)	77.8 ± 3.1	97.9 ± 11.4	80.7 ± 3.1	75.7 ± 7.9
Mg (mg kg^−1^)	849 ± 74 B	1053 ± 127 A	843 ± 74 B	746 ± 33 B
Zn (mg kg^−1^)	8.0 ± 1.7 C	10.7 ± 2.1 C	22.0 ± 3.0 A	17.3 ± 1.4 B
Se (µg kg^−1^)	39.7 ± 3.0 B	117.4 ± 17.9 A	98.0 ± 10.0 A	122.7 ± 10.5 A
Ca removal (g ha^−1^)	37,049 ± 1884 C	51,412 ± 1716 A	38,469 ± 4049 BC	45,225 ± 5070 AB
Fe removal (g ha^−1^)	1091 ± 73 B	1635 ± 205 A	1252 ± 88 AB	1454 ± 198 AB
Mg removal (g ha^−1^)	11,957 ± 1384 B	17,588 ± 2151 A	13,274 ± 1920 B	14,146 ± 555 B
Zn removal (g ha^−1^)	114 ± 27 C	179 ± 34 B	345 ± 58 A	330 ± 33 A
Se removal (mg ha^−1^)	555 ± 49 C	1957 ± 307 AB	1559 ± 263 B	2308 ± 127 A

Different letters (if any) indicate significant differences (*p* ≤ 0.05) according to LSD test. CP: crude protein; NDF: neutral detergent fiber; ADF: acid detergent fiber; ADL: acid detergent lignin; Ash: ash content; OMD: organic matter digestibility; RFV: relative forage value.

**Table 4 plants-13-01917-t004:** Distribution of the split-split plot experimental design of the study. Zn soil represents the application of 50 kg ZnSO_4_·7H_2_O ha^−1^ only at the beginning of the first campaign, while Zn foliar indicates two applications per campaign of 4 kg ZnSO_4_·7H_2_O ha^−1^, one at the beginning of tillering and the other at the appearance of the first node.

		Study year (Main plot)			
		2017/18		2018/19	
		Zn Soil Application (Subplot)		Zn Soil Application (Subplot)	
		Zn Soil	No Zn Soil	Zn Soil	No Zn Soil
Foliar application (Sub-subplot)	Zn Foliar	Zn Soil + Zn Foliar 1	No Zn Soil + Zn Foliar 1	Zn Soil + Zn Foliar 1	No Zn Soil + Zn Foliar 1
		Zn Soil + Zn Foliar 2	No Zn Soil + Zn Foliar 2	Zn Soil + Zn Foliar 2	No Zn Soil + Zn Foliar 2
		Zn Soil + Zn Foliar 3	No Zn Soil + Zn Foliar 3	Zn Soil + Zn Foliar 3	No Zn Soil + Zn Foliar 3
		Zn Soil + Zn Foliar 4	No Zn Soil + Zn Foliar 4	Zn Soil + Zn Foliar 4	No Zn Soil + Zn Foliar 4
	No Zn Foliar	Zn Soil + No Zn Foliar 1	No Zn Soil + No Zn Foliar 1	Zn Soil + No Zn Foliar 1	No Zn Soil + No Zn Foliar 1
		Zn Soil + No Zn Foliar 2	No Zn Soil + No Zn Foliar 2	Zn Soil + No Zn Foliar 2	No Zn Soil + No Zn Foliar 2
		Zn Soil + No Zn Foliar 3	No Zn Soil + No Zn Foliar 3	Zn Soil + No Zn Foliar 3	No Zn Soil + No Zn Foliar 3
		Zn Soil + No Zn Foliar 4	No Zn Soil + No Zn Foliar 4	Zn Soil + No Zn Foliar 4	No Zn Soil + No Zn Foliar 4

## Data Availability

The data supporting the findings of this study are available from the corresponding author if reasonable requests are made.

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
