# Peer review of "Soil and Foliar Zinc Biofortification of Triticale (x *Triticosecale*) under Mediterranean Conditions: Effects on Forage Yield and Quality"

_plants, 2024, doi:10.3390/plants13141917_

Round 1

Reviewer 1 Report

Comments and Suggestions for Authors

Thank you for giving me the article to review. In my opinion, however, it requires minor corrections before publication, which I point out below

1 note: not to change, but the research seems to be quite distant, because it is older than 5 years

2. please explain why there were no P markings in the soil?

3. Please change the unit of micro siemens to milli siemens

4. Please do not quote literature older than 15 years

6. It is worth giving in a table or in the form of a drawing of the experiment diagram

7. On figure 1, please note a value not only statistic

8. Which methods were used to determine nitrogen and potassium please describe in more detail

The article is written in the correct language, the introduction and abstract are sufficient

Author Response

Manuscript entitled "Soil and Foliar Zinc biofortification in Triticale (x Triticosecale) under Mediterranean conditions: effects on forage yield and quality" by “García-Latorre, Reynolds-Marzal, De la Peña-Lastra, Pinheiro and Poblaciones” submitted to Plants.

We thank the reviewers for all your suggestions and corrections. We have agreed to include most of the changes proposed by the reviewers to improve the manuscript and make it more readable. We hope that it will now be acceptable for publication in Plants. Corrected manuscript is attached with and without changes indicated by using the track changes mode in MS Word.

In the following point by point reply, the response with the justification of the changes can be found. In black are the reviewers´ comments and in bold are our responses.

Comments to reviewer 1:

Thank you for giving me the article to review. In my opinion, however, it requires minor corrections before publication, which I point out below

Comment 1 note: not to change, but the research seems to be quite distant, because it is older than 5 years

We acknowledge that the research may be a bit older than desirable. We thank the reviewer to consider that it is still worth publishing, as we also think that the results obtained in the experiments are still relevant and up to date to better understand the biofortification strategies to improve nutrient status in winter cereals, in particular, in triticale.

Comment 2. please explain why there were no P markings in the soil? Thank you very much, it was an oversight. The P olsen value of the test floor has already been added.

Comment 3. Please change the unit of micro siemens to milli siemens. This has been changed.

Comment 4. Please do not quote literature older than 15 years. Thank you for pointing this. We have replaced all the older references with literature from the last five  years, except for three of them, as those were the methods we followed and have remained unchanged since their publication.

Comment 6. It is worth giving in a table or in the form of a drawing of the experiment diagram. Thanks for the suggestion. A table summarizing has been added in the Materials and Methods section to improve clarification.

Comment 7. On figure 1, please note a value not only statistic. We have done some testing and adding numerical data worsens the clarity of the figure. As the fundamental data are added in the results and discussion we have not thought it appropriate to make this change.

Comment 8. Which methods were used to determine nitrogen and potassium please describe in more detail. The methods have been described in the text.

The article is written in the correct language, the introduction and abstract are sufficient. We want to thank the reviewer for their comments that have helped us improve our manuscript.

Reviewer 2 Report

Comments and Suggestions for Authors

Major comments:

1)Abstract: too short, does not give  necessary information

 (i)Chemical form of Zn and concentration of the salt applied should be indicated

ii)-Lack of Zn supply at the second season should be written;

(iii)- quality parameters should be indicated

Introduction:

2) There are a lot of publications devoted to Zn biofortification of cereals (see the reference list). That supposes that broader discussion of the appropriate investigations should be made in the Introduction section to reveal the main problems and reveal the novelty of the work

Results and Discussion:

3)At the end of the manuscript it is written that Zn was not supplied to soil at the second cropping year. This should be indicated at the beginning- otherwise it is difficult to understand the results. And furthermore, the differences in the results between two cropping years may be connected not only with weather but with a single application of Zn

4)  Tables 2-3: Is there a misprint?: in Table 2 in no-Zn supply Se content is about 91-97 mcg/kg while in Table 3 another value is indicated -39.7 mcg/kg – In this respect does Zn positively affect Se concentration? It seems desirable to give additional information between Se-Zn interaction both with and without Se supply on other crops

Minor comments:

Title ‘ Soil and Foliar Zinc biofortification in Triticale (x Triticosecale) under Mediterranean conditions: effects on forage yield and quality’ change to ‘Soil and Foliar Zinc biofortification of Triticale (x Triticosecale) under Mediterranean conditions: effects on forage yield and  quality’

4)Table 1- all abbreviations should be deciphered in the footnotes of the Table. Change the Title of the Table as the significance data should be indicated in the footnote. The same for other Tables. Please, add the title of the first column in all Tables

5) line 41 ‘Regarding animals, including humans…’-style, change

6) line 83 ‘ 0.30 ± 0.03 mg Zn-DTPA kg-1’ – 0.3 mg Zn per kg of soil ??? Why have you included DTPA???- You have already indicated that Zn was extracted using DTPA. I recommend to delete DTPA here

7) what is ADF?- decipher

8) line 115 ‘acid (ADF)- revise to ‘ADF’

9) line 199 ‘did not produced’ change to ‘did not produce’

10) what was the size of experimental plot

11) line 322 ‘Forage was harvested at the beginning of milky grain’- this should be also indicated in the Results and discussion sections

19) chlorophyll content is desirable

20) Conclusion line 359 ‘50 kg ha-1’ kg –that refers to Zn sulphate but not Zn

20) references:

Ref. 58- add number of article;

Ref.4,14,15,26- add pages or number of article

Comments on the Quality of English Language

no comments

Author Response

Manuscript entitled "Soil and Foliar Zinc biofortification in Triticale (x Triticosecale) under Mediterranean conditions: effects on forage yield and quality" by “García-Latorre, Reynolds-Marzal, De la Peña-Lastra, Pinheiro and Poblaciones” submitted to Plants.

We thank the reviewers for all your suggestions and corrections. We have agreed to include most of the changes proposed by the reviewers to improve the manuscript and make it more readable. We hope that it will now be acceptable for publication in Plants. Corrected manuscript is attached with and without changes indicated by using the track changes mode in MS Word.

In the following point by point reply, the response with the justification of the changes can be found. In black are the reviewers´ comments and in bold are our responses.

Reviewer 2

Comments and Suggestions for Authors

Major comments:

Comment 1)Abstract: too short, does not give  necessary information

 (i) Chemical form of Zn and concentration of the salt applied should be indicated

  1. ii) Lack of Zn supply at the second season should be written;

(iii) quality parameters should be indicated

The points addressed by the reviewer have been included in the abstract.

Introduction:

Comment 2) There are a lot of publications devoted to Zn biofortification of cereals (see the reference list). That supposes that broader discussion of the appropriate investigations should be made in the Introduction section to reveal the main problems and reveal the novelty of the work. In accordance with the comments made by the other reviewer, some of the citations used have been updated, including the relevant information from them.

Results and Discussion:

Comment 3) At the end of the manuscript it is written that Zn was not supplied to soil at the second cropping year. This should be indicated at the beginning- otherwise it is difficult to understand the results. And furthermore, the differences in the results between two cropping years may be connected not only with weather but with a single application of Zn. Mentions to the unique application of Zn to the soil in the first year have been added at the beginning of both the results and the discussion. Additionally, the differences between years have been connected to the single application of Zn.

Comment 4) Tables 2-3: Is there a misprint?: in Table 2 in no-Zn supply Se content is about 91-97 mcg/kg while in Table 3 another value is indicated -39.7 mcg/kg – In this respect does Zn positively affect Se concentration? It seems desirable to give additional information between Se-Zn interaction both with and without Se supply on other crops. We have revised the data set, and the results are correct. Table 2 does not represent the No-Zn supply for each year studied, it is the mean of all the samples collected in each year for all the parameter, which include all the Zn treatments (No-Zn, Soil Zn, Foliar Zn and S+F Zn). That is the reason why the mean is higher in Table 2 than the one observed for the No Zn observed in the Table 3. More information regarding the interaction between Zn and Se has been provided in the discussion.

Minor comments:

Comment Title ‘Soil and Foliar Zinc biofortification in Triticale (x Triticosecale) under Mediterranean conditions: effects on forage yield and quality’ change to ‘Soil and Foliar Zinc biofortification of Triticale (x Triticosecale) under Mediterranean conditions: effects on forage yield and quality’. We agree, this has been changed.

Comment 4)Table 1- all abbreviations should be deciphered in the footnotes of the Table. Change the Title of the Table as the significance data should be indicated in the footnote. The same for other Tables. Please, add the title of the first column in all Tables. The changes in tables, titles and footnotes have been included in the text following the considerations of the reviewer.

Comment 5) line 41 ‘Regarding animals, including humans…’-style, change. Thank you for the suggestion. We have rewritten the sentence in order to improve the style and avoid repetition.

Comment 6) line 83 ‘ 0.30 ± 0.03 mg Zn-DTPA kg-1’ – 0.3 mg Zn per kg of soil ??? Why have you included DTPA???- You have already indicated that Zn was extracted using DTPA. I recommend to delete DTPA here. All right, it is already corrected in the text.

Comment 7) what is ADF?- decipher. The meaning of these acronyms (ADF would be acid detergent fiber) have been made clearer in the text.

Comment 8) line 115 ‘acid (ADF)- revise to ‘ADF’. It has been revised.

Comment 9) line 199 ‘did not produced’ change to ‘did not produce’. Thank you for pointing out the error. It has been changed.

Comment 10) what was the size of experimental plot. Plot size for each treatment was 15 m2 (3 m× 5 m). It has been added to the Materials and methods section.

Comment 11) line 322 ‘Forage was harvested at the beginning of milky grain’- this should be also indicated in the Results and discussion sections. Thanks for the suggestion. This point has been addressed in the Results and the Discussion sections.

Comment 19) chlorophyll content is desirable. Thank you for the suggestion, we agree that the chlorophyll content would have been a valuable addition to the manuscript. Although we have incorporated the measurement of this parameter to our more recent experiments, unfortunately, we could not collect that information at the moment of the study.

Comment 20) Conclusion line 359 ‘50 kg ha-1’ kg –that refers to Zn sulphate but not Zn. Thanks for the comment. This point has been been clarified in the conclusions.

Comment 20) references: Ref. 58- add number of article; Thank you for this comment, as we have noticed this reference was already in the bibliography. This reference has been substituted by the correct one and the number has been added to said cite.

Ref.4,14,15,26- add pages or number of article. The information requested has been added.
